# Addressing Resource Scarcity across Sign Languages with Multilingual Pretraining and Unified-Vocabulary Datasets

**Gokul NC**[1,2]**, Manideep Ladi**[2]**, Sumit Negi**[2]**, Prem Selvaraj**[1]**,**
**Pratyush Kumar**[1,2,3]**, Mitesh Khapra**[1,2]
[1]AI4Bharat, [2]IIT Madras, [3]Microsoft Research
gokulnc@ai4bharat.org, manideepladi@gmail.com, sumitnegi662@gmail.com, prem@ai4bharat.org,
pratyush@cse.iitm.ac.in, miteshk@cse.iitm.ac.in

## Abstract

There are over 300 sign languages in the world, many of which have very limited or no labelled sign-to-text datasets. To address low-resource data scenarios, self-supervised pretraining and multilingual finetuning have been shown to be effective in natural language and speech processing. In this work, we apply these ideas to sign language recognition. We make three contributions. First, we release *SignCorpus*, a large pretraining dataset on sign languages comprising about 4.6K hours of signing data across 10 sign languages. *SignCorpus* is curated from sign language videos on the internet, filtered for data quality, and converted into sequences of pose keypoints thereby removing all personal identifiable information (PII). Second, we release *Sign2Vec*, a graph-based model with 5.2M parameters that is pretrained on *SignCorpus*. We envisage *Sign2Vec* as a multilingual large-scale pretrained model which can be fine-tuned for various sign recognition tasks across languages. Third, we create *MultiSign-ISLR* – a multilingual and label-aligned dataset of sequences of pose keypoints from 11 labelled datasets across 7 sign languages, and *MultiSign-FS* – a new finger-spelling training and test set across 7 languages. On these datasets, we fine-tune *Sign2Vec* to create multilingual isolated sign recognition models. With experiments on multiple benchmarks, we show that pretraining and multilingual transfer are effective giving significant gains over state-of-the-art results. All datasets, models, and code has been made open-source via the 🤟OpenHands toolkit[1].

## 1 Introduction

According to the World Health Organization, 500 million people in the world have disabling hearing loss, and the number is expected to go up to 700 million by 2050. Sign language is a common medium of communication amongst the Deaf and Hard of Hearing (DHH). In sign languages, expressions are through the visuospatial modality by combining movements of fingers, arms, face, and upper body. Sign languages are full-fledged languages that have evolved over long durations of time to have their unique lexical and grammatical features. Across the world, there are over 300 different sign languages which often have very little commonality with languages spoken in respective regions. Finally, there are large variations in signing between people using the same sign language. All these characteristics make it challenging to build tools to reduce communication barriers between DHH people and those who do not know sign languages.

---

[1]https://openhands.ai4bharat.org

36th Conference on Neural Information Processing Systems (NeurIPS 2022) Track on Datasets and Benchmarks.

One class of such tools are AI-based sign language recognition (SLR) models which can process a video of a person signing to identify a single character (finger-spelling recognition), a single signed unit or a gloss (isolated sign recognition), or a sequence of glosses (continuous sign recognition). As summarized by Koller [2020], there has been growing interest in creating datasets and training SLR models for various sign languages across the world. However, the available sign language datasets are still limited in various ways. First, in sheer size these datasets are orders of magnitude smaller than speech or text datasets. For example, even the largest continuous sign language corpora have up to 1,00,000 sign instances [Koller et al., 2015] while speech corpora have millions of spoken words and text corpora have billions of tokens. Second, the availability of resources is non-uniform across languages. American Sign Language has most data with over 4 datasets with a total of 57,187 videos, while Indian Sign Language has only 1 publicly available dataset with 4,287 videos [Sridhar et al., 2020]. Finally, these datasets are often not diverse with representation of only few signers and with limited variation in backgrounds and other visual conditions. In contrast, speaker diversity is an essential metric in speech datasets. In summary, the field of sign language recognition is lacking datasets to develop usable AI tools.

In text and speech processing, several techniques have been developed for low-resource languages, i.e., languages that have limited amount of data resources. One such technique is self-supervised learning to build language and acoustic models on large unlabelled datasets. Removing the requirement to label data allows curating data from web-scale resources, even for low-resource languages. The second technique is to create transfer amongst one or more high-resource language and related low-resource languages by training a single model across languages. Such multilingual models may additionally enforce linguistic transfer with aligned model outputs, such as using a common decoding script in a machine translation system. In this work, we apply these techniques to sign languages with both contributions in datasets and models trained on these datasets.

We make the following three dataset contributions. First, we create *SignCorpus* a pretraining dataset for sign language modelling across 10 sign languages - American, Australian, British, Chinese, Greek, Indian, Korean, Russian, Spanish, Turkish. *SignCorpus* comprises of about 4,568 hours of temporal sequences of 75 pose keypoints from the upper half of the body including 11 keypoints in the face. The modality of pose keypoints focuses on the movement of body parts and their relation to each other, while removing extraneous features such as the background. Further, pose keypoints do not contain personally identifiable information which may be a concern in releasing a large public dataset. The videos are curated from language specific sources on YouTube and other platforms and filtered based on a variety of identified quality measures. Second, we create *MultiSign-ISLR* a multilingual dataset with temporal sequences of pose keypoints from 11 publicly available isolated sign language recognition datasets across 7 sign languages: American, Argentinian, Chinese, German, Greek, Indian, and Turkish. *MultiSign-ISLR* also maps the datasets to have a common label set; for example, the signs for the gloss *cat* in American Sign Language are mapped to the same label as the signs for the gloss 猫 in Chinese Sign Language. In total, *MultiSign-ISLR* has more than $300K$ videos across $5,144$ aligned labels. Third, we create *MultiSign-FS* a new label-aligned multilingual dataset for finger spelling from videos we curated from YouTube and other sources. *MultiSign-FS* contains sequences of pose trajectories across 7 sign languages – American, Argentine, Chinese, Greek, German, Indian, Turkish and 69 finger spelt characters and digits.

We analyze the value of the above datasets by training and evaluating models for sign recognition. First, we pretrain a Decoupled GCN network with 5.2M parameters on *SignCorpus* with a self-supervised objective based on Dense Predictive Coding [Han et al., 2019]. We call this model *Sign2Vec*, which is a multilingual large-scale pretrained model that can be adapted for various sign recognition tasks across multiple languages. We then fine-tune *Sign2Vec* on *MultiSign-ISLR* to obtain a multilingual sign language recognition model for 7 languages. We compare the accuracy of the model against baseline models varying both pretraining and finetuning stages to be monolingual or multilingual. We see significant improvements against each of these baselines and also against state-of-the-art models trained individually for each dataset. We also finetune *Sign2Vec* on *MultiSign-FS* to create a multilingual finger spelling recognition dataset. With similar comparisons with baselines we report large improvements in accuracy both due to multilingual pretraining and joint fine-tuning. With these results, we demonstrate the value of the datasets we release - *SignCorpus*, *MultiSign-ISLR*, and *MultiSign-FS*, and effectiveness of the multilingual *Sign2Vec* model. All models are released as part of the 🤲OpenHands repository.

| Dataset | SL | GL | Vocab | Signers | Videos | Hrs |
|---|---|---|---|---|---|---|
| ASLLVD [de Amorim et al., 2019] | American | English | 2745 | 6 | 9,748 | 2.7 |
| AUTSL [Sincan and Keles, 2020] | Turkish | Türkçe | 226 | 43 | 38,336 | 20.5 |
| BosphorusSign22k [Özdemir et al., 2020] | Turkish | Türkçe | 744 | 6 | 22,542 | 19 |
| CSL [Huang et al., 2019] | Chinese | Mandarin | 500 | 50 | 125,000 | 108.8 |
| DEVISIGN [Chai et al., 2014] | Chinese | Mandarin | 2000 | 30 | 24,000 | 21.9 |
| GSL [Adaloglou et al., 2021] | Greek | Elliniká | 310 | 7 | 40,785 | 6.4 |
| INCLUDE [Sridhar et al., 2020] | Indian | English | 263 | 7 | 4,287 | 3.6 |
| LSA64 [Ronchetti et al., 2016] | Argentinian | Spanish | 64 | 10 | 3,200 | 1.9 |
| MSASL [Joze and Koller, 2019] | American | English | 1000 | 222 | 25,513 | 24.0 |
| Phoenix-W-S03 [Forster et al., 2012] | German | Deutsch | 271 | 1 | 3659 | 0.5 |
| WLASL [Li et al., 2020] | American | English | 2000 | 119 | 21,083 | 14.0 |

Table 1: The diverse set of existing ISLR datasets which we study in this work. *SL* indicates the sign language and *GL* indicates the language used for the corresponding glosses

## 2 Related Work

### 2.1 Datasets for Isolated Sign Recognition

Isolated Sign Language Recognition (ISLR) is one of the primary tasks in sign language processing, which involves classifying individual signs in a video to corresponding glosses. A gloss is a fundamental text unit of sentence in sign language, usually conveying a meaning or a concept through a single signing action. Given the importance of ISLR and ease of building datasets for this task compared to other SL tasks (like CSLR), there has been relatively more number of datasets released for ISLR. Each dataset usually consists of a subset of the vocabulary from the sign language being considered, with multiple samples for each gloss to train ML models.

Table 1 shows the list of 11 ISLR datasets (across 7 sign languages) that are openly available and being studied in this work. American and Chinese SL datasets are the largest available datasets, whereas Argentinian and German SL have the lowest amount of data. It is to be noted that none of these datasets are large enough to represent rich diversity of vocabulary used in practice. Also, there are still a large number of unexplored sign languages for which even small ISLR datasets have not yet been built. This signifies the importance of building datasets and pretrained models which can enable information transfer under few-shot or even zero-shot settings.

### 2.2 Low-resource Text and Speech Processing

Self-supervised learning is one of the key approaches in NLP to improve the performance of models using both unlabeled data as well as labeled data. This usually involves 2 phases – an initial training of the model using the unlabeled data, commonly known as pretraining phase. The pretrained model is then used for other downstream tasks by further training on different task-specific objectives, and called fine-tuning phase. This method was particularly effective for the BERT model by Devlin et al. [2019], wherein a large Transformer-encoder model was trained on a large corpus of unlabelled text, and then fine-tuned on language understanding tasks. Since then, there has been improved self-supervised techniques not only for NLP [Doddapaneni et al., 2021], but also for other domains like speech [Schneider et al., 2019], image [Dosovitskiy et al., 2020] and video [Feichtenhofer et al., 2022] including action recognition [Han et al., 2019].

Furthermore in NLP, training a single model for multiple languages has enabled information transfer from related high-resource languages to low-resource languages [Chau and Smith, 2021]. This multilingual transfer can also happen at the pretraining stage: Self-supervised learning is effective during fine-tuning even for languages that are unseen during pretraining stage, showing that such language models are good few-shot learners [Winata et al., 2021]. This effectiveness of multilingual transfer suggests that modeling human language using large amounts of unlabelled data captures patterns that generalize across languages.

In addition to multilingual models, converting text data from languages using different scripts to a common representation has shown to improve transfer to low-resource languages, for tasks like Machine Translation [Ramesh et al., 20d] and Automatic Speech Recognition [Javed et al., 2022]. Such script unification makes explicit the transfer of common knowledge like shared formal

vocabulary across related languages. In the domain of speech synthesis, it is now a common approach to unify all languages to the International Phonetic Alphabet (IPA) when training multilingual text-to-speech (TTS) models [Zhang et al., 2019].

## 2.3 Self-supervised Training for Sign Languages

There has been limited research on improving performance on low-resource sign languages, especially given that almost all sign languages listed in Table 1 are low-resource. This was brought to the notice of the wider NLP community in a recent paper by Yin et al. [2021]. Self-supervised training for sign languages has been recently studied by Selvaraj et al. [2022] and Hu et al. [2021]. In the work by Hu et al. [2021], the pretraining is performed only on the hand skeleton and thus does not capture facial features or other bodily articulations. Also, no pretraining dataset is created for SL, but rather they reuse the pose data available from hand skeleton datasets, which is not in any way related to sign language data. In contrast, we pretrain on a large unlabeled corpus that contains clean sign language pose data covering different SLs. In our previous work [Selvaraj et al., 2022], the training was done only on Indian Sign Language (ISL), while this work expands the training to 10 sign languages and increases the dataset size by 4×. As we show, this increased diversity of pretraining results in improved accuracy.

## 3 *SignCorpus*: Pretraining Dataset across 10 Sign Languages

The success of pretraining in NLP and speech is due to large corpora such as the OSCAR corpus [Abadji et al., 2022] which contains multiple terabytes of text collected for 166 languages. In the following, we describe how we use similar web resources to build *SignCorpus*.

### 3.1 Curating Content

First, we identify the sign languages that we consider in *SignCorpus*. Based on analysis of the amount of content available in YouTube, we decided to focus on 10 sign languages - American, Australian, British, Chinese, Green, Indian, Korean, Russian, Spanish, and Turkish. For Indian Sign Language (ISL), in our earlier work we presented a pretraining corpus of 1,129 hours [Selvaraj et al., 2022]. Hence we focus on the remaining 9 languages in this work.

Second, we identify a list of channels and content sources for each sign language. This was performed by searching for related keywords both in English and in native languages. We found that a majority of the sources were from news and media outlets who provided sign language interpreters. The second major source was religious content which also came with sign language interpretation. A third major source of content was sign language learning material such as glossaries and tutorials. These three sources accounted for 90.7% of the videos that we found. The full list of all channels and sources, for each language, is provided in the Appendix.

We divided the videos into three categories: (a) isolated - has a single gloss signed, (b) continuous - has sequences of glosses translating a continuous stream of audio, and (c) multiple isolated - has a few glosses signed one-by-one usually with a pause (for teaching purposes). All three types of videos are of relevance to pretraining which is done over small windows of time of the order of a couple of seconds. The statistics of the number of videos and their length found across each type and sign language is shown in Table 2. In total, we found 84,420 videos totalling to 4,145 hours of content across the 9 sign languages. Majority of the content was in American Sign Language, while Turkish SL has the least.

### 3.2 Dataset Processing

First, we crop videos to capture the signer. We manually find the coordinates to crop for each video and then use FFmpeg for cropping. An example of this is shown in the Appendix. In our experience, these crop coordinates remain relatively constant throughout all frames in a video and often throughout different videos in the same channel/playlist.

As the second and crucial step, we convert the RGB video frames into frames of pose keypoints. An example conversion is shown in Figure 1. This is a crucial choice we make because of three reasons. One, all personally identifiable information (PII) is removed thereby eliminating most

| Sign Language | Isolated | | Continuous | | Multiple Isolated | | Total Raw Data | | Total Data |
| | Videos | Hours | Videos | Hours | Videos | Hours | Videos | Hours | Hours |
| --- | --- | --- | --- | --- | --- | --- | --- | --- | --- |
| American SL | 34,510 | 53.05 | 12,271 | 1,104.25 | - | - | 46,781 | 1,157.30 | 879.25 |
| Australian SL | - | - | 1,035 | 117.70 | 89 | 2.20 | 1,124 | 119.90 | 71.99 |
| British SL | 198 | 0.93 | 3,102 | 808.85 | 726 | 16.93 | 4,026 | 826.71 | 675.78 |
| Chinese SL | 8,588 | 17.50 | 947 | 333.20 | 16 | 2.00 | 9,551 | 352.70 | 305.65 |
| Greek SL | - | - | 3,773 | 524.25 | 37 | 2.00 | 3,810 | 526.25 | 475.18 |
| Korean SL | 2,171 | 7 | 4,980 | 446.50 | 102 | 1.00 | 7,253 | 454.50 | 426.72 |
| Russian SL | 7,633 | 7.72 | 1,359 | 451.05 | 254 | 17.73 | 9,246 | 476.50 | 412.67 |
| Spanish SL | - | - | 1,514 | 171.63 | 141 | 6.83 | 1,655 | 178.46 | 161.48 |
| Turkish SL | 671 | 0.85 | 268 | 38.35 | 35 | 14 | 974 | 53.20 | 48.50 |
| Indian SL | - | - | - | 1,129.00 | - | - | - | 1,129.00 | 1,129.00 |
| Total | 53,771 | 87.05 | 29,249 | 5124.78 | 1400 | 62.69 | 84,420 | 5274.52 | 4586.22 |

Table 2: Category-wise statistics of the unlabeled dataset that we collect in our work. The last column indicates the final number of hours per sign language after all pre-processing and quality checks.

privacy concerns. This is of particularly important because large pretraining corpora for audio or text have significantly lesser risks of PII content. Two, accurate pose extractors have been trained on much larger datasets. This modularizes the sign recognition problem instead of framing it as an end-to-end video-to-gloss classifier. Three, the sign language recognition models working on pose data have one or two orders of magnitude fewer parameters than models that process video directly.

We use the MediaPipe Holistic [Grishchenko and Bazarevsky, 2020] toolkit to extract pose keypoints. The library provides separate models for body pose, facial, and hand landmark extraction. Across these three, 543 landmarks (33 body pose, 21 left hand, 21 right hand, and 468 face) are extracted. Out of these points, we store 75 keypoints - covering both hands, body, and face. A detailed description of all keypoints, the ones we have chosen, and the data format are in the Appendix. Thus, for all 363.32M frames in the dataset, we use MediaPipe to extract 75 keypoints represented by x and y coordinates which are relative to the width and height of the cropped video.

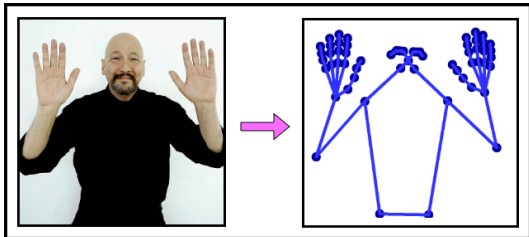

Figure 1: Illustration for image to pose keypoints conversion. Left: RGB image of a person signing. Right: 75 keypoints which we use to represent the frame.

### 3.3 Filtering Videos based on Quality Check

Given the large scale of data, we designed and implemented multiple data quality checks. First we remove all intervals of time wherein no person is detected for a heuristically chosen interval length of 3 seconds. Second, we remove all videos where multiple people are visible in the crop of the signer. With these two rules, we filter out 6.5% of the curated videos.

Next, we apply frame-wise checks and statistical filtering based on those checks. The first check was to ensure that a person is clearly visible in the cropped frame. This ruled out videos in which pose extraction fails, a person is not present, or those which only contain hands of signers. The second check was to ensure that a person is signing, and not just speaking. We did this by checking that both hands are visible and are being moved. The third check was to ensure that the person is oriented properly facing the camera. We applied these binary checks to individual frames and aggregated an average score across all frames in a video. We then identified thresholds for each check to be met by a video. These thresholds were identified language-wise and based on the type of video - isolated, continuous, or multiple isolated. Applying these thresholds resulted in varying ratios of accepted videos - ranging from 87% for British Sign Language channels to 99% for Korean Sign Language

channels. Details of the identified thresholds, and the fractions of videos meeting different checks by language are in the Appendix.

Upon completing all quality checks, the number of hours of data by language is shown in the last column of Table 2. In combination with the data we released for Indian Sign Language (ISL) earlier [Selvaraj et al., 2022], we have a total of 4,586 hours of pretraining data in *SignCorpus*.

## 4   *Sign2Vec*: Model Pretrained on *SignCorpus*

In this section, we describe the network architecture and pretraining approach that utilizes *SignCorpus* dataset. We call this pretrained model *Sign2Vec* as we envision it as a multilingual large-scale model that produces a semantic vector representation for a given sign video.

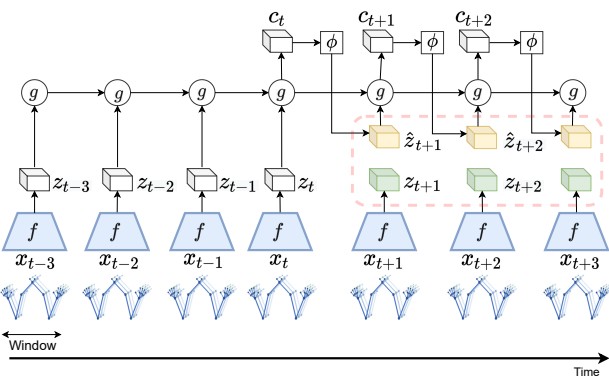

Figure 2: Model architecture in *Sign2Vec* for DPC pretraining as proposed by Selvaraj et al. [2022]

As a backbone to train *Sign2Vec*, we choose a graph convolution network (GCN) based architecture. For pose-based tasks GCN models are common, as pose keypoints can be considered as a graph of nodes on which the convolution operation models the spatial relationship between connected keypoints. In the case of isolated sign recognition where we have a sequence of pose frames, we also need to model the temporal relation between keypoints. To this end, Spatio-Temporal convolutions on 3D graphs are used, which are referred to as ST-GCNs [Lin et al., 2020]. In our work, we specifically choose Sign Language GCN as the backbone network, which was proposed by Jiang et al. [2021] for modeling sign language pose representations. SL-GCN combines STC-attention with a model called Decoupled-GCN [Cheng et al., 2020] from action recognition research and extends it to isolated sign recognition. Decoupled GCN enables increased neural capacity while adding only few additional parameters. We also include an attention-guided dropout mechanism called DropGraph [Cheng et al., 2020], which avoids overfitting even when using many GCN layers.

When training a model on a labelled sign recognition task, an SL-GCN can directly be used to classify between glosses. For pretraining, we need additional network components. We follow the self-supervised technique called Dense Predictive Coding (DPC) [Han et al., 2019]. DPC has been shown to be effective in capturing the temporal relations due to three key features. First, DPC does not predict at the frame-level but instead at feature-level where the features represent compressed and generalized latent representations of a segment of video. Second, DPC represents temporal information across short time ranges instead of individual or neighbouring frames with high stochastic variation. Third, DPC learns high-level semantics of future time frames instead of predicting masked low-level representations, thereby avoiding modelling simple interpolation.

We now describe the model architecture with the SL-GCN backbone and support for DPC as shown in Figure 2. The individual pose-frames $p_k$ from a video are grouped into non-overlapping contiguous windows. Each such window $x_i$ spanning $N$ frames is then passed to the shared SL-GCN network denoted as $f$, which produces an embedding $z_i$. These embeddings are passed sequentially to multiple layers of Gated Recurrent Unit [Cho et al., 2014]. This GRU network $g$ is responsible for modeling the context-dependent representation $c_i$. The training objective is to predict the representation of next set of timesteps $t + 1, t + 2, \ldots$ given the contextual representation $c_t$. More precisely, we predict the contextless spatio-temporal embeddings $\hat{z}_{t+1}, \hat{z}_{t+2}, \ldots$ of the future windows $x_{t+1}, x_{t+2}, \ldots$ by

passing the context vector $c_t, c_{t+1}, \ldots$ through a shared fully-connected layer $\phi$ auto-regressively at each step. We also generate the actual embeddings $z_{t+1}, z_{t+2}, \ldots$ using the windows $x_{t+1}, x_{t+2}, \ldots$ across the entire training batch. Then finally for each future timestep starting from $t + 1$, we use a variant of the loss function *Noise-Contrastive Estimation* which maximizes the lower-bound of mutual information between relevant samples, called InfoNCE loss [Oord et al., 2018]. Specifically, for a given predicted representation, say $\hat{z}_{t+1}$, we consider the actual representation $z_{t+1}$ as the positive sample, and use other representations from the same sequence (like $z_{t+2}$) as the temporal negative samples & samples from other sequences in the same batch as approximate spatial negative samples. We then use the contrastive objective to minimize the distance between the positive samples in the embedding space, and maximize that of negative samples.

In our implementation, we define a window as consisting of $N = 10$ frames. We use 10 layers of GCN and the entire SL-GCN denoted $f$ has 4.9M parameters. The embedding generated denoted by $z$ is a vector of 256 floating point numbers. The function $g$, implemented by GRUs consists of 0.25M parameters. The function $\phi$ is implemented by a fully-connected layer with 0.1M parameters. More details about the model implementation can be found in the Appendix.

## 5 Label-Aligned Multilingual Finetuning Dataset

In this section, we describe the contributions that we make in releasing labeled datasets. Specifically, we explain the label-aligned datasets that we release for existing isolated sign recognition datasets of different sign languages as well as new finger-spelling datasets that we curate.

### 5.1 *MultiSign-ISLR*: Vocabulary-Unified Multilingual ISLR Dataset

As shown in Table 1, we explore 11 different ISLR datasets across 7 sign languages. The languages in which glosses for these datasets are written are based on the lingua franca of the respective countries where those sign languages are standardized. As seen in the table, the written languages of the labels of these datasets vary significantly. For a multilingual model trained on these datasets, there would be more than 10K labels to predict. The larger number of labels not only makes classification harder but also makes transfer between languages harder with distinct labels for each sign language. To address this, we unify the glosses of all sign languages to a common representation. We choose English as the common language, as it is widely spoken and because there are around 6K glosses in English (in American and Indian SLs). First, we go through all the label sets in English and perform a few normalization operations to convert them to a consistent format. For instance, we normalize "Thank you" to "THANKS" for consistency. For the remaining 4K non-English glosses, we initially translate them all to English using Google Translate. Then we manually verify if the translations are correct by cross-checking against an online dictionary (Wiktionary) and performing corrections wherever necessary. Finally, we combine all the glosses into a unique vocabulary set. This process reduces the number of labels from 10,123 (actual representation) to 5,144 (unified representation), almost a $2\times$ reduction. For instance, labels implying "dog" (English) in different languages like "köpek" (Turkish), "hund" (German), "perro" (Spanish), etc. were mapped to a single standard gloss "DOG". We release these normalized and label-aligned glosses for all the 11 datasets studied in this work through the

👐OpenHands library. We also explain in the Appendix the set of all standard rules that we follow for normalization of glosses and cross-lingual alignment. In addition, for each of the datasets we create sequences of pose keypoints with MediaPipe as already described in Section 3.2. In the next section, we demonstrate the utility of this unified multilingual dataset by training models based on this data.

### 5.2 *MultiSign-FS*: Multilingual Finger-Spelling Datasets

Another sign recognition problem is to identify individual characters in finger spelling (FS). To the best of our knowledge, there is no existing label aligned multilingual dataset on FS. We create such a dataset across 7 sign languages (the same ones as in the previous section) and call it *MultiSign-FS*. Similar to *SignCorpus*, videos for *MultiSign-FS* are primarily collected from YouTube[2]. We manually label each video for finger-spelt regions, crop them as short videos, and extract pose keypoints from these videos. The label set for each language consists of all alphabets in the language and the 10

---

[2]In addition, we also collect a few videos from specific websites like SpreadTheSign and Gebärden lernen for the languages that have inadequate data on YouTube.

| Sign Language | Alphabets | Numerals | Total Signs | Signers per class | Train-set size | Test-set size |
|---|---|---|---|---|---|---|
| American | 26 | 10 | 36 | 15.6 | 428 | 134 |
| Argentine | 27 | 10 | 37 | 4.0 | 111 | 37 |
| Chinese | 30 | 10 | 40 | 13.1 | 414 | 112 |
| Greek | 24 | 10 | 34 | 3.9 | 100 | 34 |
| German | 26 | 10 | 36 | 4.6 | 132 | 36 |
| Indian | 26 | 10 | 36 | 4.7 | 123 | 49 |
| Turkish | 32 | 10 | 42 | 3.9 | 122 | 42 |
| Total Unique | 59 | 10 | 69 | - | 1430 | 444 |

Table 3: Details of the finger-spelling datasets we curate in *MultiSign-FS*

numerals $(0 - 9)$. The alphabet size varies across languages, although most of them predominantly overlap with the English-Roman alphabet. In the case of Chinese, we use the more commonly used Pinyin Roman representation. The detailed alphabet list for each language is shown in the Appendix. On average, we have 5 different signers for each finger spelled character. We split the dataset into train and test sets roughly in the ratio of 4:1 while ensuring it is signer independent, i.e., videos in the test set are signed by different signers than in the train set. The statistics of *MultiSign-FS* are shown in Table 3. Upon unifying the character-set across all languages, there are 69 unique characters including 10 numerals. Through the 🤟OpenHands library, we release all the character-sets along with the extracted poses from the collected videos for all the 7 languages.

## 6  Experimental Results

In this section, we present results of pretraining on *SignCorpus* dataset, followed by fine-tuning for both the label-aligned datasets *MultiSign-ISLR* and *MultiSign-FS*.

Following the network architecture and self-supervised training strategy described in Section 4, we pretrain the SL-GCN model using the *SignCorpus* dataset. From the videos in *SignCorpus*, we sample random clips of 70 frames mapping to time-intervals of about 3 seconds. To construct a training batch, such clips are taken from different videos in *SignCorpus*. With a window of size 10 frames, we construct 7 windows from each sampled clip. Out of these, 4 windows are modelling the context vector which then predict the represenations for the subsequent 3 windows in future steps. We use a batch size of $128$ and a learning rate of $10^{-4}$ using Adam Optimizer [Kingma and Ba, 2015] for stochastic gradient descent. The training set consists of entire *SignCorpus*, whereas sampled clips from *MultiSign-ISLR* are used as a validation set to monitor the progress of training and for early-stopping. We pretrain the model for 1.2M iterations.

We now report results of fine-tuning *Sign2Vec* for *MultiSign-ISLR* and *MultiSign-FS* datasets. We study two options for fine-tuning: (a) unilingual (UFT) where for each language we fine-tune only the training set of that language, and (b) multilingual (MFT) where we fine-tune jointly for all languages with all training sets combined. When training the MFT model, we encode the input sign language as a One Hot Encoding (OHE) vector. The hyperparameters for all experiments are reported in the Appendix. We also report baseline results on fine-tuning the model pretrained only on Indian Sign Languages [Selvaraj et al., 2022] which we denote ISL. We also report state-of-the-art results reported for each of the individual benchmarks.

For *MultiSign-ISLR*, we report results in Table 4. We make three observations. First, the benefit of a larger and more diverse pretraining dataset is revealed by comparing ISL and UFT, which have the same finetuning setup but differ in the pretraining corpus. On average, UFT has 3.5% higher accuracy across datasets. The improvement is more significant for datasets with low accuracy such as ASLLVD, MSASL, and WLASL. Notably, the accuracy on INCLUDE which is an ISL dataset slightly improves even when diversifying the pretraining beyond ISL. Second, the benefit of aligned multilingual dataset is revealed by comparing UFT and MFT. On average, MFT has significantly higher accuracy by about 10%. In composition, multilingual pretraining and fine-tuning result in 13.4% improvements over results with ISL. Finally, we compare the SOTA results against MFT.

| Dataset | SOTA | ISL | UFT | MFT |
|---|---|---|---|---|
| ASLLVD | 16.5 [de Amorim et al., 2019] | 27.7 | 32.5 | **50.1** |
| AUTSL | **95.0** [an, 2021] | 88.3 | 92.1 | 91.4 |
| BosphorusSign22k | **94.9** [Gökçe et al., 2020] | 88.3 | 87.2 | 93.6 |
| DEVISIGN | 63.9 [Selvaraj et al., 2022] | 59.4 | 62.2 | **79.3** |
| INCLUDE | 93.5 [Selvaraj et al., 2022] | 94.7 | 94.9 | **96.3** |
| LSA64 | **97.8** [Selvaraj et al., 2022] | 96.3 | 97.5 | 95.9 |
| MSASL (1000) | 59.8 [Hu et al., 2021] | 18.2 | 34.0 | **67.4** |
| PHEONIX-W-S03 | - | 47.7 | **48.7** | 48.5 |
| WLASL | **47.5** [Hu et al., 2021] | 27.4 | 31.0 | 46.6 |
| Average (without Pheonix) | 71.1 | 62.5 | 66.4 | **77.6** |
| Average (with Pheonix) | - | 60.9 | 64.4 | 74.3 |

Table 4: Accuracy on *MultiSign-ISLR* dataset. ISL: Model pretrained on ISL data [Selvaraj et al., 2022] and finetuned individually. UFT: Model pretrained on *SignCorpus* and finetuned individually. MFT: Model pretrained on *SignCorpus* and finetuned jointly across all languages.

There is an average increase of 6.5% across datasets, excluding Pheonix. We note that SOTA results sometimes train on validation sets [an, 2021] and use ensemble models [Gökçe et al., 2020], which we did not perform for MFT. Further, this advantage of MFT is despite it being a single multilingual model with just 5.1M parameters while SOTA results are usually on larger models individually optimized for each dataset.

For *MultiSign-FS* we report results in Table 5. Since this is a new dataset, we create a baseline by training SL-GCN model from scratch individually on all datasets. For the other two models we report results only on multilingual finetuning, either using the model already pretrained on ISL data by Selvaraj et al. [2022] or by pretraining on *SignCorpus*. The results show 0.5% increase (SL-GCN vs ISL) in average accuracy by pretraining on a single language and a further improvement of 8% (ISL vs SignCorpus) by pretraining on multiple languages. For both models, details of the training setup and hyperparamters are in the Appendix.

| Sign Language | SL-GCN | ISL | SignCorpus |
|---|---|---|---|
| American | 73.9 | 65.7 | 76.9 |
| Argentine | 62.2 | 54.1 | 64.9 |
| Chinese | 52.7 | 43.8 | 57.1 |
| German | 63.9 | 72.2 | 75.0 |
| Greek | 20.6 | 32.4 | 38.2 |
| Indian | 38.8 | 42.8 | 44.9 |
| Turkish | 42.9 | 47.6 | 57.1 |
| Average | 50.71 | 51.2 | 59.2 |

Table 5: Accuracy on the *MultiSign-FS* dataset. SL-GCN models finetuned from scratch for each dataset. ISL: Pretrained on ISL data [Selvaraj et al., 2022] and finetuned jointly. SignCorpus: Pretrained on SignCorpus and finetuned jointly.

# 7 Conclusions

To address the low resource nature of sign language datasets, we contributed three multilingual datasets: *SignCorpus*, *MultiSign-ISLR*, and *MultiSign-FS*. Results on public benchmarks across 7 languages demonstrated effectiveness of both pretraining on a diverse multilingual dataset and joint fine-tuning on multilingual label-aligned datasets. The pretrained and multilingual finetuned models are also effective resources which can be finetuned for other sign recognition tasks and languages. More generally, our efforts in data collection and model building strongly suggest effective strategies of improving accuracy of sign language tools both for considered languages which have publicly available small datasets and for the many other languages which have far fewer resources.

### 7.1 Limitations

One of the major limitations of our work include not being able to have a human evaluation setup for the varied sign languages studied in our work. Our work does not involve manual verification of the extracted pose videos to check if native signers are able to transcribe these keypoints sequences to labels, as a proxy measure of assessing the integrity of RGB video to poses conversion. Also, with respect to the vocabulary-unification performed in Section 5.1, we do not claim that the alignment is 100% accurate since none of the authors in this work are native speakers (or signers) of those non-English languages

With respect to self-supervised learning, we believe a lot more work needs to be done to explore other pretraining strategies, and extensive benchmarking of different SSL methodologies is lacking and essential to propose what concretely works in all cases. For example, in our work, we find that accuracies do not improve upon pretraining for datasets like CSL [Huang et al., 2019] and GSL [Adaloglou et al., 2021], for which we do not report any improvements in Table 4. We believe that the pretraining method explored in this work is quite preliminary in nature, merely intended to demonstrate the utility of large-scale unlabeled data for multilingual ISLR. Also, the results and findings from this paper need to be extended and evaluated for continuous sign recognition, especially the pretraining strategy. Although there are also works like BOBSL [Albanie et al., 2021] and How2Sign [Duarte et al., 2021] which focus on aligning the sign language data from videos and audio subtitles for creating an automated corpus to train CSLR models, our currently work focuses only on collecting unlabeled videos for self-supervised pretraining.

## Acknowledgments and Disclosure of Funding

We would like to extend our immense gratitude to Microsoft's *AI4Accessibility* program (via Microsoft Philanthropies India) for granting us the compute required to carry out all the experiments in this work, through Microsoft Azure cloud platform. Our extended gratitude also goes to Zenodo, who helped us with hosting some of our large datasets [NC and Selvaraj, 2021] [Ladi et al., 2022]. Finally, we thank all the content creators and ISLR dataset curators without whose data this work would have been impossible. (Table 6 in appendix credits the list of all sources from which we created our pretraining datasets, and Table 12 credits the sources for creating the fingerspelling datasets)

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
