# OpenReview forum: "Addressing Resource Scarcity across Sign Languages with Multilingual Pretraining and Unified-Vocabulary Datasets"
_NeurIPS.cc/2022/Track/Datasets_and_Benchmarks — NeurIPS 2022 Datasets and Benchmarks _

### Official Review · Reviewer_W4d2 · 2022-07-23
**Set of multilingual sign language databases and pretrained models with significant limitations**

**Rating:** 6
**Confidence:** 5

**Strengths:**

The sequences of pose keypoints for multilingual videos are the greatest contribution of this work. The developed models, including pretrained or fine-tuned models, would interest some researchers.

**Weaknesses:**

Most sequences of pose keypoints for multilingual videos were created in previous authors' work (Selvaraj et al.; doi: 10.18653/v1/2022.acl-long.150). The developed models, including pretrained or fine-tuned models, would be of inferior interest to the research community as most researchers develop their own models and there is no consensus on the way the neural network for sign language recognition should look like. The lack of video information would confine the users of the dataset to a set of keypoints produced MediaPipe Holistic software. The usage of software allows for maintaining the anonymity of the signers. However, it is not clear whether authors were granted permission to use and process videos crawled from Internet sources. The recent research is more focused on continuous sign language recognition models. This subject is neglected in this submission, limiting the usability of the models. Since the accuracy of some models is presented, the submission could be treated as a benchmark for new models, but it should be described differently in this case.

**Additional Feedback:**

The dataset is not balanced. This makes its usage for training and evaluation of different models more complex. The implications of the structure of the datasets are not discussed in the paper.  Also, it would be nice to see the addition of which dataset in the fine-tuning is the most contributing. And the investigation of why the MFT model decreased the accuracy (Table 4) in some cases requires further attention. However, taking into account above suggestions would transform the dataset into benchmark submission, which, in my opinion, would be more interesting for researchers, as the proposed pretrained models would be rarely used by other researchers and the benchmark already has such potential.

**Clarity:**

The introduction of the dataset is quite chaotic and poorly laid out as the number of videos, justification of the number of used languages, the addition of finger-spelling, data origin, extraction, and models are introduced revealing many details that at first hinder the justification and purpose of the dataset. The paper requires proofreading. Some abbreviations are used before the first introduction (e.g., PII in Abstract).

**Correctness:**

Yes, the submission is correct. The submission is a dataset and it contains necessary information. However, since the accuracy of some models is presented without detailed comparison and an explicitly defined data division into training and testing subsets, its usage as a benchmark requires further work.

**Documentation:**

Data collection details are not revealed in sufficient detail. Readers should look at (Selvaraj et al.; doi: 10.18653/v1/2022.acl-long.150), where a detailed figure “Pipeline used to collect and process Indian SL corpus for self-supervised pretraining” is presented. It seems that the same data collection protocol was used here. The maintenance plan is not shown.

**Ethics:**

It is not clear whether permission to use videos for pose extraction was needed or granted.

**Relation To Prior Work:**

It is written that “in [Selvaraj et al., 2022], the training was done only on Indian Sign Language (ISL), while this work expands the training to 10 sign languages and increases the dataset size by 4×. As we show, this increased diversity of pretraining results in improved accuracy.” However, this is not a research paper in which neural network models are introduced but a dataset paper. And, looking at the content of the [Selvaraj et al., 2022], it seems that most efforts with data curation, crawling, and labeling were put while preparing previous work. Here, some additional languages and finger-spelling datasets are new, which cannot be seen as a novel contribution but rather an extension of the existing and published dataset. It is not true that “For Indian Sign Language (ISL), in our earlier work we presented a pretraining corpus of 1,129 hours [Selvaraj et al., 2022].149 Hence we focus on the remaining 9 languages in this work.” since in previous work results for several (other) languages are shown, indicating that the pose data for them was prepared for experiments. A clear distinction between what was done previously and where is not shown.



**Summary And Contributions:**

In this work, several sign multilingual datasets are introduced, described, and modeled: 1) SignCorpus – dataset with ten languages represented by sequences of pose keypoints for SignCorpus 2) Sign2Vect – model pretrained on SignCorpus, 3) MultiSign-ISLR - dataset with pose sequences of seven languages, 4) MUltiSign-FS- finger-spelling dataset of seven languages, 5) fine-tuned Sign2Vect models on 7 languages. The submission refers to OpenHands library, developed and presented at 60th Annual Meeting of the Association for Computational Linguistics (Selvaraj et al.; doi: 10.18653/v1/2022.acl-long.150), and it seems that most work that would be interesting for sign language – related researchers, which is related to the dataset preparation, including carving the data, labeling, and determination of pose keypoints, has been already done while preparing the paper of Selvaraj et al. The prepared models are of lesser importance since many neural network-based solutions are introduced each year; hence, the usage of pretrained Sign2Vect would be limited in the next few years. The recent research is more focused on continuous sign language recognition models. This subject is neglected in this submission, limiting the usability of the models.

---

> ### Author Response · Authors · 2022-08-13
> **Part 1 -- Responses to the comments & feedback of Reviewer W4d2**
>
> We thank you for taking the time to review our work. Please find our responses below.
>
> > _Contribution_: The submission refers to OpenHands library, developed and presented at 60th Annual Meeting of the ACL, and it seems that most work that would be interesting for sign language – related researchers, which is related to the dataset preparation, including carving the data, labeling, and determination of pose keypoints, has been already done while preparing that paper.
> > _Weakness_: Most sequences of pose keypoints for multilingual videos were created in previous authors' work
> > _Relation to Prior Work_: looking at the content of the [Selvaraj et al., 2022], it seems that most efforts with data curation, crawling, and labeling were put while preparing previous work. Here, some additional languages and finger-spelling datasets are new, which cannot be seen as a novel contribution but rather an extension of the existing and published dataset.
> > _Relation to Prior Work_: ...in previous work results for several (other) languages are shown, indicating that the pose data for them was prepared for experiments. A clear distinction between what was done previously and where is not shown.
>
> We strongly disagree with the characterization that there is no major contribution wrt previous work Selvaraj et al, ACL 2022. Our work makes the following novel contributions -
>
> 1. The prior work presents an unlabelled dataset for Indian Sign Language alone. In this work, we create an unlabelled dataset, SignCorpus, for 10 sign languages. This involved significant curation of language-specific sources of data. Also, we propose manual and automated approaches to grade the quality of data from a source and to filter data based on it. No other unlabelled dataset supports this diversity of sign languages.
>
> 2. We also release a unified vocabulary set for 11 different ISLR datasets to enable benchmarking of multilingual sign recognition. No other such dataset exists.
>
> 3. We also create new fingerspelling datasets for 7 different sign languages. No other such dataset exists.
>
> 4. We study multilingual sign language models where a single model performs inference across different languages, as benchmarked on the label-aligned multilingual dataset we release. Prior work does not study this.
>
> 5. We create and release Sign2Vec trained on the largest unlabelled sign corpus across 10 languages. We believe that this model can be finetuned for downstream tasks across these languages. We also demonstrate that such multilingual pretraining also enables multilingual fine-tuning, creating a unified model.
>
> 6. We release fingerspelling and isolated sign recognition models for 7 different sign languages. Fingerspelling models are not considered in the mentioned prior work. For those sign languages for which the prior work supports isolated sign recognition, we show increases in accuracy. Specifically, the average accuracy increases by 6.5% across 6 sign languages.
>
> > _Contribution_: The prepared models are of lesser importance since many neural network-based solutions are introduced each year; hence, the usage of pretrained Sign2Vect would be limited in the next few years.
> > _Weakness_: The developed models, including pretrained or fine-tuned models, would be of inferior interest to the research community as most researchers develop their own models and there is no consensus on the way the neural network for sign language recognition should look like.
>
> We strongly disagree with this statement. The lack of standardized datasets in the low-resource domain of sign languages indicates that development of AI resources for sign language is still in its early stages. Indeed, in this stage of research evolution there would be competing ideas which over time would lead to multiple attempts. We believe it is uncharitable or even incorrect to dismiss one or more of such as being “inferior” just due to the existence of other solutions. We also note that the models released in this paper, which is submitted to a datasets track, also provide extrinsic value of the datasets created. Training models with and without the datasets show improvements in accuracy.

---

> > ### Author Response · Authors · 2022-08-13
> > **Part 2 -- Responses to the comments & feedback of Reviewer W4d2**
> >
> > > _Weakness_: The usage of pose allows for maintaining the anonymity of the signers. However, it is not clear whether authors were granted permission to use and process videos crawled from Internet sources.
> > > _Ethics_: It is not clear whether permission to use videos for pose extraction was needed or granted.
> >
> > Creating datasets from public sources of data, such as websites (eg. Common-Crawl) or Youtube (eg. UCF 11) and releasing for research purposes is common practice. However, being conscious of ethical considerations, we have made a few choices. First, as discussed in the paper, by converting videos to pose trajectories we remove PII of signers’ faces. Second, we source data only from channels which contain public and general-purpose knowledge, such as news broadcasts, datasets for learning sign language, etc. Third, in the paper, we include the full list of the individual channels from which we collect data. This is to both acknowledge the valuable contribution of these channels and also to enable data users to flag a specific channel which can be removed by us. As we rightly be conscious of ethical issues in data collection, we as a community must also be mindful of our responsibility in creating AI resources for low-resourced domains such as sign language.
> >
> > > _Correctness_: since the accuracy of some models is presented without detailed comparison and an explicitly defined data division into training and testing subsets, its usage as a benchmark requires further work.
> >
> > We rebut this characterization of lack of correctness. When reporting results on any existing benchmark, such as the ISLR benchmarks in 7 languages, which are duly cited in Table 1, we use the test dataset as proposed in the original paper. Also, we compare the results of our work with previous work as detailed in Table 4 with due citations. For the benchmarks we release on fingerspelling, we detail test and train splits in Table 3.
> >
> > > _Weakness_: The lack of video information would confine the users of the dataset to a set of keypoints produced by MediaPipe Holistic software.
> >
> > Yes we make the choice to release only keypoint data due to ethical considerations on removing PII. We use the open-sourced MediaPipe Holistic model as you note, though we believe that any alternative model, such as OpenPose, can also be adapted to this procedure. For users or researchers who would like to work with the original videos, we provide the full list of video URLs in the metadata files of our released dataset, along with the list of all video channels in the Appendix.
> >
> > > _Documentation_: Data collection details are not revealed in sufficient detail. Readers should look at (Selvaraj et al.), where a detailed figure “Pipeline used to collect and process Indian SL corpus for self-supervised pretraining” is presented. It seems that the same data collection protocol was used here.
> >
> > The section A of the Appendix had explained in detail (for around 6 pages) about the data collection process adopted in this paper (not same as previous work), followed by a pictorial representation of the extended pipeline in Figure 9 at the end of the appendix.

---

> > > ### Author Response · Authors · 2022-08-24
> > > **Follow-up with Reviewer W4d2**
> > >
> > > Thank you again for your reviews. We hope that we have addressed all your queries and provided clarifications. If there are any further questions or comments, please do let us know. We will be glad to address them, or incorporate suggestions into the paper. If there are no more concerns and if you are clear & optimistic about the objectives & contributions of our work, it would be great if you can reconsider your rating of the paper, at your own discretion.

---

> > > > ### Comment · Reviewer_W4d2 · 2022-08-26
> > > > **Still not clarified**
> > > >
> > > > Thank you for the response. The answers clarified some issues. However, the most important one that addresses potentially the greatest contribution of this work is still valid.
> > > >
> > > > You have answered that “The prior work presents an unlabelled dataset for Indian Sign Language alone. “ and after reading your previous work (Selvaraj et al.; doi: 10.18653/v1/2022.acl-long.150; https://aclanthology.org/2022.acl-long.150.pdf), I cannot agree with such a statement. In Selvaraj et al., you have written as follows: “With a view to cover a diverse set of languages, we study 7 different datasets across 6 sign languages as summarized in Table 1. For each of these datasets, we generate pose-based data using the Mediapipe pose estimation pipeline (Grishchenko and Bazarevsky,2020), which enables real-time inference in comparison with models such as OpenPose (Cao et al.,2018).” Figure 1 in both works is shared as the acquisition process is the same. Therefore, as I wrote before, it seems that most work that would be interesting for sign language–related researchers, which is related to the dataset preparation, including carving the data, labeling, and determination of pose keypoints, has been already done while preparing the paper of Selvaraj et al.  Table 1 in that work contains a list of processed datasets from six sign languages (AUTSL, CSL, DEVISIGN, GSL, INCLUDE, LSA64, and WLASL). Also, Table 2 contains the results of some models trained on OpenHands model and tested on multiple sign languages. This means that they were already prepared. Some new languages were added but others were rejected from the dataset presented in the reviewed paper (e.g., Armenian?). Why? Which datasets are newly added and which are extended? What is the amount of extension in the number of signs and other data characteristics? This would indicate the amount of new work. Note that you have written in previous paper that “We release pose-based datasets and 4 different ISLR models across 6 sign languages.”  Hence, it cannot be stated now that “The prior work presents an unlabelled dataset for Indian Sign Language alone. “
> > > >
> > > > A solid explanation and clarification of what is done here in relation to the previous authors’ work are still needed.

---

> > > > > ### Author Response · Authors · 2022-08-27
> > > > > **Clarifications regarding relation to previous work**
> > > > >
> > > > > We thank you for engaging in this discussion and providing another opportunity to clarify the key differences to the prior work ([Selvaraj et al., ACL 2022](https://aclanthology.org/2022.acl-long.150/)).
> > > > >
> > > > > As you have noted, both works are interested in the following research question – Can we improve the accuracy of AI systems for isolated sign language recognition with pretraining strategies? The prior work answered the question in affirmative by showing that we can pretrain a model on a large corpus of Indian Sign Language pose data and then fine-tune it for various languages showing improvements on existing isolated sign recognition baselines across languages.
> > > > >
> > > > > In this work, we extend this further along the following axes -
> > > > >
> > > > > 1. We propose **multilingual pretraining**. While the prior work, only curated and pretrained on one language - Indian Sign Language, in this work we curate diverse sources of data across 9 sign languages. This is an important contribution, and indeed took a considerable fraction of our research time, since we looked for language specific sign language channels spanning a wide range of cultural variations and our own lack of knowledge in these languages. Further, given the diversity of the videos, we had to devise new methods to ensure data quality. We demonstrate that such multilingual pretraining is beneficial with improvements in accuracy across the same set of monolingual benchmarks as considered in the prior work.
> > > > > We may consider a parallel in the world of NLP where initial efforts to create language models for single languages (eg. BERT) were followed by efforts to create multilingual language models (eg. mBERT).
> > > > >
> > > > > 2. We propose **multilingual finetuning**. While the prior work considered finetuning for various languages, each time a new finetuned model was created. In this work, we proposed finetuning a single multilingual model. To this end, we proposed a method for creating a common label set (which we release) and then trained a single multilingual model. We showed significant improvements in accuracy due to this multilingual finetuning.
> > > > > Again, a parallel in the NLP world is the creation of single multilingual models such as multilingual translation models like M2M-100.
> > > > >
> > > > > 3. We also study **fingerspelling**. The prior work only looked at isolated signs, whereas we create entirely new benchmarks for fingerspelling, align labels, and report baselines on these models.
> > > > > Again, a parallel in the NLP world is the creation of multiple NLU tasks which can be used to test a pretrained language model.
> > > > >
> > > > > None of the above – multilingual pretraining, multilingual finetuning, and fingerspelling – is considered in the prior work. Along these three axes we make several datasets contributions – SignCorpus, MultiSignISLR, and MultiSignFS, and establish baselines on these datasets with several models which improve on the SOTA accuracy in many cases.
> > > > >
> > > > > We hope this clarifies some of the concerns, please let us know if there are any further questions.
> > > > >
> > > > > Regarding the following question:
> > > > >
> > > > > > *Which datasets are newly added and which are extended? What is the amount of extension in the number of signs and other data characteristics? This would indicate the amount of new work.*
> > > > >
> > > > > We will revise the paper to indicate in table-1 marking which ISLR datasets were already used in previous work, and which datasets are being used anew now (which is already evident by comparing with table-1 of previous work). But please note that this cannot be a measure of “amount of new work” done – these are the already existing ISLR benchmark datasets for which we study techniques to improve performance (like self-supervised learning and vocabulary unification).

---

> > > > > > ### Comment · Reviewer_W4d2 · 2022-08-29
> > > > > > **Updated score**
> > > > > >
> > > > > > The provided discussion directs the attention to the already established view on the new things and lacks a direct and clear explanation of the cited issue from the first answer to my review. I believe I can't get more.
> > > > > > In my opinion, the paper is good enough to be published. I hope that the readers of your previous work would not have difficulties distinguishing the old dataset from the new dataset to replicate the results.
> > > > > > I've updated my score for the paper.

---

### Official Review · Reviewer_5g1u · 2022-07-24
**Necessary and timely benchmark**

**Rating:** 9
**Confidence:** 4

**Strengths:**

The released data is large scale and the benchmark unifies the sign recognition task across multiple signed languages. The benchmark is necessary and timely.


**Weaknesses:**

The key weakness of the submission is the lack of human toplines for native signers for each of the 10 languages considered attempting the task on the pose information alone, which is what models have to work with. In particular, the extracted poses may be noisy or inaccurate, and so we need to know whether a native signer is able to translate the extracted poses released as the dataset into the target local language gloss / identify the SL sign.

A methodology question: For "MultiSign-ISLR" what about signs that don't have a clear, single meaning in the common label set? Are these thrown out? Problem not acknowledged?

A metholdogy question: Section 3.3 filtering does not explicitly check that videos labeled with a particular sign all actually depict that sign being surfaced. In automated scrapes of data, it is exceedingly easy to get noisy productions of neighbor signs or signs embedded in phrases. Can the authors expand on how they ensured that each labeled video depicted the sign in isolation (when necessary) and that all labeled videos are in fact depicting the same underlying sign?


**Additional Feedback:**

"Sign2Vec as a multilingual foundation model" the reviewer requests the authors to re-consider using the term "foundation model" in one of two ways. Either (1) cite the Stanford paper that establishes this term or (2) avoid using the term because it is a Stanford power-grab to make the "Stanford Center for Foundation Models" synonymous with large-scale pretraining, which was not invented at Stanford. The reviewer favors option (2), but if you're going to call your model a "foundation model", cite the paper that introduces the terminology, because it is non-standard. Sign2Vec is a large-scale, multimodal pretrained model, which you can also be proud of, mouthfull-tho-it-may-be.

Typo line 38 "1,00,000 signs"

Line 75 and elsewhere citation should be parenthetical but appears as noun. Line 132 (and elsewhere) opposite problem, citation appears parenthetically that should be compiled as a noun.

Super weird spacing around the numbers in line 271; maybe just drop the commas in these.


**Clarity:**

There are some typos but the core arguments and presentation of the paper are clear.


**Correctness:**

The experiments are complete and include both existing state of the art sign rec models and a new pretrained model based on the large, released corpus.


**Documentation:**

Yes, the landing site is easy to navigate and clear.


**Ethics:**

The submission includes some footnotes on limitations, but these could be promoted to main paper content (e.g., authors not being fluent in all 10 of the target SLs or local spoken languages is reasonable but it -is- also a limitation).


**Relation To Prior Work:**

Yes, the work is well-positioned.


**Summary And Contributions:**

This submission introduces a multilingual benchmark of signed languages, including both local spoken language glosses and a unified multi-SL to English gloss annotation for single-language translation/decoding. The benchmark includes both fingerspelling (boring, but more common and easy; usually not multilingual!) and sign recognition (cooler, harder, usually not multilingual!) /and/ sequential signing (hardest, realest, usually not multilingual). The benchmark tasks are sign recognition (cool!) and fingerspelling (yawn, but we "get it").

---

> ### Author Response · Authors · 2022-08-13
> **Responses to the comments & feedback of Reviewer 5g1u**
>
> We thank you for taking the time to review our paper. Please find our responses below.
>
> > The key weakness of the submission is the lack of human toplines for native signers for each of the 10 languages considered attempting the task on the pose information alone, which is what models have to work with. In particular, the extracted poses may be noisy or inaccurate, and so we need to know whether a native signer is able to translate the extracted poses released as the dataset into the target local language gloss / identify the SL sign.
>
> We agree with your assessment. Human baselines are required for establishing both top-lines and bottom-lines. On the one hand, as is standard practice, we need human baselines on the labelled datasets we release to establish what human-level performance is. Secondly, as you mention, and as is less common, we need to identify if there is significant information loss when moving from data-rich videos to data-sparse pose trajectories. We believe these are valid open questions, that were beyond the scope of this Dataset paper.
>
> > A methodology question: For "MultiSign-ISLR" what about signs that don't have a clear, single meaning in the common label set? Are these thrown out? Problem not acknowledged?
>
> We create the common label set dynamically, i.e., if a new label is encountered for a sign, we append it to the list. Also, most ISLR datasets have labels with unambiguous meaning. In the few cases where there are multiple interpretations, the context is also clearly mentioned in labels. For example, in a Chinese Sign Language dataset, we found that there were different signs for “aunt”, which were properly annotated as “aunt (paternal)” and “aunt (maternal)” in Mandarin.
>
> > A methodology question: Section 3.3 filtering does not explicitly check that videos labeled with a particular sign all actually depict that sign being surfaced. In automated scrapes of data, it is exceedingly easy to get noisy productions of neighbor signs or signs embedded in phrases. Can the authors expand on how they ensured that each labeled video depicted the sign in isolation (when necessary) and that all labeled videos are in fact depicting the same underlying sign?
>
> Our primary contribution is the pretraining dataset, SignCorpus, which is unlabelled. So, we do not encounter the challenge you mention about segmenting videos into individual signs. For the labelled datasets, we only source videos which are known to contain individual signs. Other related work however has made progress in addressing the problem of sign segmentation, in particular the creation of the [BOBSL dataset](https://www.robots.ox.ac.uk/~vgg/data/bobsl/).
>
> > Clarity: There are some typos but the core arguments and presentation of the paper are clear.
> > Typo 1: line 38 -- "1,00,000 signs"
> > Typo 2: Line 75 and elsewhere citation should be parenthetical but appears as noun. Line 132 (and elsewhere) opposite problem, citation appears parenthetically that should be compiled as a noun.
> > Typo 3: Super weird spacing around the numbers in line 271; maybe just drop the commas in these.
>
> We thank you for pointing out these errors. They have been fixed in the paper.
>
> > Limitation: The submission includes some footnotes on limitations, but these could be promoted to main paper content (e.g., authors not being fluent in all 10 of the target SLs or local spoken languages is reasonable but it -is- also a limitation).
>
> We agree. We have moved them to the limitations section.
>
> > Feedback: "Sign2Vec as a multilingual foundation model" the reviewer requests the authors to re-consider using the term "foundation model" in one of two ways. Either (1) cite the Stanford paper that establishes this term or (2) avoid using the term because it is a Stanford power-grab to make the "Stanford Center for Foundation Models" synonymous with large-scale pretraining, which was not invented at Stanford. The reviewer favors option (2), but if you're going to call your model a "foundation model", cite the paper that introduces the terminology, because it is non-standard. Sign2Vec is a large-scale, multimodal pretrained model, which you can also be proud of, mouthfull-tho-it-may-be.
>
> Thank you for this comment. We did some additional reading about the history of the term foundation model. Based on this, we have decided to avoid the term and use the more explicit description of the models we build.

---

> > ### Comment · Reviewer_5g1u · 2022-08-15
> > **Human toplines**
> >
> > Thanks for your response! I'm a bit confused by the bit about human toplines yet you said both that
> >
> > > Human baselines are required for establishing both top-lines and bottom-lines. On the one hand, as is standard practice, we need human
> > baselines on the labelled datasets we release to establish what human-level performance is.
> >
> > That is, establishing human-level performance is standard practice when releasing a labeled dataset.
> >
> > > We need to identify if there is significant information loss when moving from data-rich videos to data-sparse pose trajectories. We believe these are valid open questions, that were beyond the scope of this Dataset paper.
> >
> > That is, establishing human-level performance is beyond the scope of a dataset paper.
> >
> > I disagree that human performance is outside the scope of a dataset release, especially in a track designed for datasets and benchmarks specifically. If we can't be sure the pose extraction preserves enough information for a native signer to identify the sign being surfaced, it's possible the dataset has large chunks of value-less video information or noisy evaluation / training data.
> >
> > Minimally, the lack of human evaluations should feature heavily in the Limitations section, since the authors do not include fluent signers of many of the languages in the dataset to be released. To not include a human topline of people in the community the dataset includes and targets is erasure of the that community (e.g., fluent, deaf signers of the studied signed languages).

---

> > > ### Author Response · Authors · 2022-08-24
> > > **Response regarding human toplines**
> > >
> > > We thank you for your response. We are in agreement that human baselines are required, as we have mentioned. When we stated that the creation of such baselines was outside the scope of the work, we were referring to the limitation of not being able to create a human evaluation setup for the varied sign languages studied in our work. Yes, we agree this should be listed as a limitation and we will do just that.

---

> > > > ### Comment · Reviewer_5g1u · 2022-08-25
> > > > **human toplines**
> > > >
> > > > Sounds good! Thanks for clarifying again.
> > > >
> > > > Also, for everyone/the AC, just to be clear: I think this paper is great and valuable, I just wanted to make sure the human topline thing makes it into the Limitations discussion and to understand where the authors stand on it. Hence, my rating of the paper stays at 9 (i.e., high).

---

### Official Review · Reviewer_bUDd · 2022-07-27
**Unified Vocabulary Dataset for Sign Languages**

**Rating:** 7
**Confidence:** 5
**Correctness:** The paper is technically sound
**Clarity:** The paper is well written

**Strengths:**

1. The proposed dataset is of great interest to the community.
2. The proposed technique can improve the state of several downstream tasks.
3. The paper is well written, and the data collection technique is clearly mentioned

**Weaknesses:**

No major weaknesses.
The paper should cite BOBSL and How2Sign datasets too.

**Additional Feedback:**

None

**Documentation:**

The dataset is well documented

**Ethics:**

No major concerns

**Relation To Prior Work:**

The paper covers most of the related work. But there is no mention on How2Sign and BOBSL.

**Summary And Contributions:**

The authors propose a new sign language dataset that contains sign language videos in different languages. The authors also propose a novel network,  Sign2Vec, which is a large-scale pretrained model for sign language tasks.

---

> ### Author Response · Authors · 2022-08-13
> **Responses to the comments & feedback of Reviewer bUDd**
>
> We thank you for taking the time to review our paper. Please find our responses below.
>
> > Weakness: The paper should cite BOBSL and How2Sign datasets too.
> > Relation to Prior Work: But there is no mention on How2Sign and BOBSL.
>
> Thanks for your suggestion. These citations were missing since our paper is on isolated sign language recognition, while both How2Sign and BOBSL focus on continuous sign language. However, upon reflection, we felt that they share a similar spirit of collecting large corpora of data from publicly available datasets and building sign language models. We have thus added the citations.

---

> > ### Author Response · Authors · 2022-08-24
> > **Follow-up with Reviewer bUDd**
> >
> > Thank you again for your reviews. We hope that we have addressed all your queries and provided clarifications. If there are any further questions or comments, please do let us know. We will be glad to address them, or incorporate suggestions into the paper. If there are no more concerns and if you are clear & optimistic about the objectives & contributions of our work, it would be great if you can reconsider your rating of the paper (if possible), at your own discretion.

---

### Official Review · Reviewer_7e3q · 2022-07-28
**A hard, bold and much needed contribution for Natural Language Processing on Sign Language.**

**Rating:** 9
**Confidence:** 4
**Clarity:** The paper is clearly written, well ex…

**Strengths:**

The data gathering procedure is well described and justified, using techniques inspired of other tasks, datasets and models that have similar obstacles, such as low resource.
The model architecture of the Sign2Vec pre-trained model is well described and justified.
The experimental results show realistic improvements on multiple benchmarks and competitive results on the others.
Given that the task is difficult, understudied and it's medium and representation is demanding, we find this bold try a success.

**Weaknesses:**

As the authors mention, there are still many other pretraining strategies and model architectures to be tested.

**Additional Feedback:**



**Correctness:**

The claims in the papers are well justified and explained; the step descriptions are correctly specified and detailed and the experiments are reasonable.

**Documentation:**

The authors specify that the datasets models and code is available and open sourced.

**Ethics:**

The authors use publicly available videos on the Youtube platform to build their dataset. The videos are transformed into pose keypoints for multiple reasons, one of which is to remove personally identifiable information from the context of the video. Although commendable to remove personal information from the medium, the authors should also think about personal information from the signed language content. The ethics concern is limited (but not null) since these videos are freely available online and the video's domains are selected and should contain minimal PII (~90% of it is mainly news, sign language tutorials and religious content).

**Relation To Prior Work:**

Related work and prior work from the authors and other teams is clearly presented, discussed and referenced.

**Summary And Contributions:**

A hard, bold and much needed contribution for Natural Language Processing on Sign Language.
The authors

---

> ### Author Response · Authors · 2022-08-13
> **Responses to the comments & feedback of Reviewer 7e3q**
>
> We thank you for taking the time to review our paper. Please find our responses below.
>
> > Weakness: As the authors mention, there are still many other pretraining strategies and model architectures to be tested.
>
> Yes, we agree that there are multiple pretraining strategies. Our primary contribution, being part of the Datasets track of NeurIPS, was to create the unlabelled dataset for pretraining, and small labelled datasets for downstream tasks. We used a pretraining strategy from the open-source to provide a first baseline. But we believe that a comprehensive study of different pretraining strategies will be a valuable contribution to this community and other adjacent areas such as activity detection.
>
> > Ethics: Although commendable to remove personal information from the medium, the authors should also think about personal information from the signed language content. The ethics concern is limited (but not null) since these videos are freely available online and the video's domains are selected and should contain minimal PII (~90% of it is mainly news, sign language tutorials and religious content).
>
> We thank you for raising this issue. We were conscious of various ways in which PII could leak into the data. We have made some specific choices. First, as you mention, we do not include the videos of the signers. Then, we made the conscious choice to only include data from sources which are in the general public information domain, often news and learning content. All domains we have taken the data from are listed in Table 8 of the appendix. We are not aware of any source which could contain personal information, such as a video log or individual conversations. Nevertheless, we are open to receiving inputs on any source violating our constraints, and would remove them from the public datasets.

---

> > ### Author Response · Authors · 2022-08-24
> > **Follow-up with Reviewer 7e3q**
> >
> > Thank you again for your reviews. We hope that we have addressed all your queries and provided clarifications. If there are any further questions or comments, please do let us know. We will be glad to address them, or incorporate suggestions into the paper.

---

> > > ### Comment · Reviewer_7e3q · 2022-08-26
> > > **Thank you for the replies**
> > >
> > > I appreciate the replies to my comments. You have indeed provide good clarifications for my concerns.

---

### Official Review · Reviewer_TzBW · 2022-07-28

**Rating:** 6
**Confidence:** 4
**Correctness:** Yes.
**Clarity:** Yes, well written.

**Strengths:**

1. The dataset contributions are sufficient.
2. Useful for the group of the Deaf and Hard of Hearing (DHH) and positive social implications.

**Weaknesses:**


1. The datasets are only partly uploaded.
2. Lack some manual evaluations for dataset quality.

**Additional Feedback:**

1. Suggest adding some manual evaluations for dataset quality, such as the accuracy of image to pose keypoints conversion

**Documentation:**

Only part datasets are uploaded.

**Ethics:**

No.

**Relation To Prior Work:**

Yes, clearly discussed.

**Summary And Contributions:**

This paper focuses on resource scarcity across sign languages and makes the following datasets contributions:
- Release a multilingual sign language dataset SignCorpus comprising about 4.6K hours of signing data across 10 sign languages.
- Create a multilingual and label-aligned dataset MultiSign-ISLR of sequences of pose keypoints from 14 labelled datasets across 7 sign languages.
- Create a new finger-spelling dataset MultiSign-FS across 7 sign languages.

Additionally, the authors release a pre-trained sign language model Sign2Vec pre-trained with the dataset SignCorpus for downstream sign language tasks and demonstrate its effectiveness on MultiSign-ISLR and MultiSign-FS.

---

> ### Author Response · Authors · 2022-08-13
> **Responses to the comments & feedback of Reviewer TzBW**
>
> We thank you for taking the time to review our paper. Please find our responses below.
>
> > Weakness 1: The datasets are only partly uploaded.
> > Documentation: Only part datasets are uploaded.
>
>  The smaller datasets were all public, but there was a technical issue in uploading the largest dataset (~350GB). They have been uploaded fully now in the same link, and openly available since the beginning of August.
>
> > Weakness 2: Lack some manual evaluations for dataset quality.
> > Feedback: Suggest adding some manual evaluations for dataset quality, such as the accuracy of image to pose keypoints conversion
>
> Dataset quality is important, especially given the variety of sources of data. We dedicate Section 3.3 in the paper to explain the manual evaluations done, including building calibrated heuristics to filter out data perceived to be of lower quality. This is further detailed in Appendix A. Specific to your question about pose keypoint conversion, Table 9 in the appendix shows the overall quality of pose data for each body-part during the conversion.

---

> > ### Author Response · Authors · 2022-08-24
> > **Follow-up with Reviewer TzBW**
> >
> > Thank you again for your reviews. We hope that we have addressed all your queries and provided clarifications. If there are any further questions or comments, please do let us know. We will be glad to address them, or incorporate suggestions into the paper. If there are no more concerns and if you are clear & optimistic about the objectives & contributions of our work, it would be great if you can reconsider your rating of the paper (if possible), at your own discretion.

---

### Official Review · Reviewer_UD6b · 2022-07-28
**Important sign language dataset!**

**Rating:** 9
**Confidence:** 4
**Correctness:** The claims made by the authors seem c…

**Strengths:**

(1) SignCorpus is the largest ever collected sign language pretraining dataset. As the authors show this is very useful for finetuning neural network models for multilingual sign identification and finger spelling. I think there are plenty of other applications for such a pretraining dataset, for ex: for unsupervised MT into sign languages or even simply for augmenting sign language training with generated signs/poses.
(2) Well explained data collection process and solid experiment results for benchmarking.


**Weaknesses:**

The only weakness I see is that a lot of manual effort went into annotating the dataset by the authors/researchers themselves. Now that they have decent ML models for sign language prediction,  I wonder if this can be used to speed up the data collection process itself (some kind of life-long learning paradigm). Another related question is how large can such a dataset be given that sign language speakers are a minority. Are there enough youtube videos out there to build larger models? We may need to use different approaches depending on the answer to this.

**Additional Feedback:**

N/A

**Clarity:**

The paper is well written and structured (even the appendix).


**Documentation:**

The paper itself does not contain links to the dataset and needs an update. But the links provided in the openreview submissions page seem to be working now and contain all the datasets.

**Ethics:**

Youtube videos are converted to skeletal poses before being used/released. So this is inline with privacy requirements.

One small concern I have is about the nameless sign-language youtubers and content creators who are indirectly helping further this research into sign-languages. I understand youtube’s standard licenses do not require this. But some kind of acknowledgement to their work would provide positive support to a minority community. Perhaps a full list of contributors/channels on the dataset website or a paragraph within the paper summarizing their help.


**Relation To Prior Work:**

Introduction and related work sections provide a good summary of prior works including datasets, models and results.


**Summary And Contributions:**

This paper provides three sign language datasets which by itself is already useful (sign languages are largely under-resourced/under-studied). The authors also provide in-depth analysis of the extraction process, data attributes and a few baseline accuracy numbers using a graph-convolution network. Overall the paper is very well structured, easy to follow and critical for sign language ML research in my opinion.

---

> ### Author Response · Authors · 2022-08-13
> **Responses to the comments & feedback of Reviewer UD6b**
>
> We thank you for taking the time to review our paper. Please find our responses below.
>
> > Weakness 1: The only weakness I see is that a lot of manual effort went into annotating the dataset by the authors/researchers themselves. Now that they have decent ML models for sign language prediction, I wonder if this can be used to speed up the data collection process itself (some kind of life-long learning paradigm).
>
> Most of our dataset collection was automated and did not require manual effort. Only the MultiSign-FS and MultiSign-ISLR datasets required manual annotation. The former for collecting finger-spelling datasets, and the latter for mapping glosses from different languages to a common English vocabulary. The main dataset that we released, called SignCorpus, is a large unlabeled dataset used for self-supervised learning, which did not involve anything manual. However, it involved quality checking using a small subsets of data from different channels to establish the quality of different sources. Yes, as sign language models get more accurate, we believe creating would get more automated.
>
> > Weakness 2: Another related question is how large can such a dataset be given that sign language speakers are a minority. Are there enough youtube videos out there to build larger models? We may need to use different approaches depending on the answer to this.
>
> This is a valid concern, and analogous to scarcity in data of under-represented natural languages. In our search for sign language videos, we spent quite some time to identify sources for the 5-6K hours of data across 10 languages. However, there is an expectation that this may change for the better. Lot more content such as news broadcasts, tech events are being created along with signer videos. Also, governments of several countries have mandated to include signed content along with spoken content.
>
> > Documentation: The paper itself does not contain links to the dataset and needs an update.
>
> We included the link to the dataset in the “Dataset URL” section of the submission. This was as required by this track of NeurIPS. The link mentioned in the abstract also contains different sections dedicated for different datasets that we release.
>
> > Ethics: One small concern I have is about the nameless sign-language youtubers and content creators who are indirectly helping further this research into sign-languages. I understand youtube’s standard licenses do not require this. But some kind of acknowledgement to their work would provide positive support to a minority community. Perhaps a full list of contributors/channels on the dataset website or a paragraph within the paper summarizing their help.
>
> We are in full agreement. We have now listed all YouTube channels & playlists from which the videos were collected for each of the 10 sign languages in the appendix. We have also thanked all contributors in the acknowledgement section.

---

> > ### Author Response · Authors · 2022-08-24
> > **Follow-up with Reviewer UD6b**
> >
> > Thank you again for your reviews. We hope that we have addressed all your queries and provided clarifications. If there are any further questions or comments, please do let us know. We will be glad to address them, or incorporate suggestions into the paper.

---

### Comment · Reviewer_CSEe · 2022-08-29
**brief, last-minute ethics review**

Apologies for both the brevity and the last-minute nature of this review.

First and foremost, I applaud the authors for this resource. Working with minoritized and under-resourced communities always introduces ethical challenges that can be avoided by sticking to well-trodden ground, and unless more researchers take up the challenges (as these authors have done) to contend with the difficulties implicit in working with scanter sources and challenging power dynamics, the degree to which these communities are ignored and under-resourced -- or even shut out of public imagination -- will only grow. It's great to see work that tackles those challenges so thoughtfully and responsibly.

It's worth noting that acknowledging the YouTube creators (which is the just and responsible thing to do, and will give the dataset further credibility as being for the community rather than an extraction project) also makes those creators slightly more vulnerable to de-anonymization. This is mentioned not as a problem that the authors can or should solve -- as noted, I think that acknowledging those creators is the right thing to do, and there is no halfway version that extends recognition to these creators without making them available for a different kind of recognition. Rather, I want to highlight this as a balancing concern for future developments or iterations of this project; there might be an instance in which the risks to creators outweighs the benefit of being acknowledged.

---

### Meta-Review · Area_Chair_ZQwi · 2022-09-09

**Recommendation:** Accept
**Confidence:** 4

**Metareview:**

This paper presents a significant collection of unlabelled signing data (4.6K hours) across 10 sign languages, which have been pre-processed and converted to pose-keypoints to remove identifiable information. The paper also presents Multi-ISLR a multilingual dataset with label-alignment extracted from 11 other labelled datasets for 7 sign languages. Furthermore, the paper also provides a multilingual model, Sign2Vec, pre-trained on the unlabelled data, and fine-tuned on the labelled data, which shows the SOTA results for known tasks. Lastly, the paper also introduces a dataset for fingerspelling across 7 languages. It's likely that this work will pave the way for new research on multilingual signed language tasks.

**pros:**
* The paper provides a significant expansion for unlabelled and labelled signed data (previous work focused only on Indian Signed Languages)
* The data gathering procedure is well documented and justified
* The experimental results provide evidence of the value of the data.


**cons**
* For the SignCorpus there is a lack of a human verification process, that assesses the quality of the pose keypoints; i.e. whether they can be used by native signers to extract the target gloss.

---

### Decision · Program_Chairs · 2022-09-16

Accept